# Learning Augmentation Policies from A Model Zoo for Time Series Forecasting

## Abstract

Time series forecasting models typically rely on a fixed-size training set and treat all data uniformly, which may not effectively capture the specific patterns present in more challenging training samples. To address this issue, we introduce AutoTSAug, a learnable data augmentation method based on reinforcement learning. Our approach begins with an empirical analysis to determine which parts of the training data should be augmented. Specifically, we identify the so-called *marginal samples* by considering the prediction diversity across a set of pretrained forecasting models. Next, we propose using variational masked autoencoders as the augmentation model and applying the REINFORCE algorithm to transform the marginal samples into new data. The goal of this generative model is not only to mimic the distribution of real data but also to reduce the variance of prediction errors across the model zoo. By augmenting the marginal samples with a learnable policy, AutoTSAug substantially improves forecasting performance, advancing the prior art in this field with minimal additional computational cost.

## 1 Introduction

The time series forecasting problem has a wide range of applications, and previous literature has extensively explored this task from various perspectives. Recent deep learning approaches, particularly the powerful Transformer model (Vaswani et al., 2017), along with its subsequent models such as AutoFormer (Wu et al., 2021), PatchTST (Nie et al., 2023a), and iTransFormer (Liu et al., 2024), have shown remarkable capabilities in time series forecasting tasks. However, similar to previous learning-based time series forecasting models, these powerful frameworks are typically limited by the availability of training data. Current augmentation methods for time series data primarily rely on fixed-form preprocessing and lack effective (Wen et al., 2021; Cheung & Yeung, 2020), learnable data augmentation techniques for high-quality datasets. Since time series forecasting methods are commonly used in online systems with non-stationary streaming data that presents evolving distributions, we believe that developing effective data augmentation techniques is essential for improving the prediction accuracy and generalization ability of existing models.

Nevertheless, implementing fixed-form data augmentation techniques without clear guidance can introduce additional noise and lead to increased computational costs. Although learning-based data augmentation has significantly advanced the field of computer vision (Cubuk et al., 2019; Shorten & Khoshgoftaar, 2019), it remains under-explored in the context of time series forecasting. In this work, we propose a novel method named AutoTSAug (see Figure 1), which adaptively identifies time series data that can largely benefit from augmentation and automatically searches for optimal augmentation policies based on these samples using reinforcement learning.

The fundamental idea of our approach is to leverage a zoo of pretrained forecasting models, which are readily available in the current deep learning landscape, to establish the data filtering criterion as well as the reinforcement learning objectives. This is inspired by our empirical findings below. When we analyze the fitting errors of multiple models, we can divide the training samples into two main groups: (i) data exhibiting high diversity in prediction errors across the model zoo, and (ii) data with low diversity in prediction errors across the model zoo. An interesting observation is that models trained on the low-diversity subset typically outperform those trained on the high-diversity subset when evaluated on the same test set. We provide detailed evidence for this result in Section 3.

Figure 1: The fundamental idea of our approach involves: (i) identifying marginal data from the entire training set by assessing the prediction diversity within a zoo of pretrained models, and (ii) learning an augmentation model to transform the marginal data through reinforcement learning.

Why is this the case? We believe that when the prediction results from the model zoo are consistent, it may indicate two scenarios: either the training samples are too simple or they are too difficult for the forecasting task. Conversely, when the predictions on the model zoo are diverse, it suggests that the training process on these data (which we refer to as "*marginal samples*") is less stable. Intuitively, the most practical way to improve the training performance is to ensure that it maintains a great learning effect on the easy samples while also improving stability on the marginal samples. Notably, we can find similar learning principles in another field of curriculum learning. Therefore, a straightforward idea is that augmenting marginal samples is more likely to improve model performance, as they offer higher "marginal benefits[1]". Accordingly, the first contribution of our method is leveraging the model zoo as a diversity criterion to identify marginal samples.

Given the identified marginal samples, the next step is to augment the training set using them as anchor points. Revisiting our preliminary findings discussed above, a straightforward idea emerges: since we can achieve better performance by training the model on low-diversity data, why not transform the marginal samples by incorporating patterns that are more conducive to training while preserving the unique patterns of the marginal data? In other words, we aim for the augmented data to yield more consistent prediction errors on the model zoo. However, because the variance calculated from backtesting the augmented data across multiple forecasting models is not differentiable for the augmentation model (formulated as neural networks), we specifically employ the REINFORCE algorithm (Williams, 1992) to optimize the augmentation policy. This approach allows us to use the variance from the model zoo as the reward function while treating the input tokens of the augmentation model, which takes the form of a variational masked autoencoder (He et al., 2022), as the actions to be optimized.

The contributions of this paper are as follows:

- We introduce a novel method for learning data augmentation policies for time series forecasting. Our first contribution involves identifying marginal samples by backtesting the training set across a model zoo and using them as anchor points for augmentation. The key intuition is that improving training stability on these marginal samples is crucial for enhancing overall performance.

- The core of our method is to train a generative sequence model to predict suitable transformations of marginal samples through a REINFORCE algorithm. Unlike existing approaches that focus on generating realistic data points, our method automatically searches for optimal data augmentation policies aimed at minimizing the variance of prediction errors across the model zoo.

- Our approach shows significant improvements over common heuristic data augmentation methods across a wide range of time series datasets, demonstrating that augmenting marginal samples is a more effective and time-efficient strategy for enhancing model performance.

---

[1]Marginal benefit is a concept in microeconomics that describes the additional total revenue generated by increasing product sales by one unit. Here, we use it as an analogy to describe the performance gains from augmenting data by one unit.

## 2 RELATED WORK

### 2.1 TIME SERIES FORECASTING MODELS

With the remarkable advancements in natural language processing, various sequence-to-sequence methods have emerged to address time series forecasting (Zhang & Yan, 2022; Wang et al., 2022; Cao et al., 2024; Torres et al., 2021). Recurrent neural networks (RNNs) (Che et al., 2018; Sagheer & Kotb, 2019) are widely utilized in this area. However, due to their sequential architecture, RNNs often struggle with long-term time series forecasting. The Transformer-based models (Li et al., 2019; Wu et al., 2021; Zhou et al., 2021; Liu et al., 2021; Zhou et al., 2022; Zeng et al., 2023) have also demonstrated superior performance in long-term time series forecasting. For instance, Autoformer (Wu et al., 2021) enhances the Transformer by employing a deep decomposition architecture that progressively separates the trend and seasonal components throughout the forecasting process. PatchTST (Nie et al., 2023b) vectorizes time series data into patches of a specified size, encoding the resulting sequence of vectors through a Transformer that outputs forecasts of the desired length using an appropriate prediction head. LSTF-Linear (Zeng et al., 2023) was developed to simplify complex time series forecasting problems (Yi et al., 2024) and outperforms many leading Transformers by utilizing a set of remarkably simple one-layer linear models. iTransformer (Liu et al., 2024) adopts components with inverted dimensions and a modified architecture, demonstrating superior performance on multivariate time series data. Considering both efficiency and accuracy, we select iTransformer as the primary base model for our work and use iTransformer, PatchTST, Autoformer, and the original Transformer to construct the so-called model zoo. It is worth noting that recent literature has introduced large foundation models specifically designed for time series forecasting (Das et al., 2024; Jin et al., 2024; Bian et al., 2024). While incorporating these models into the model zoo for our data augmentation method would be an intriguing avenue for exploration, it is not the focus of this work.

### 2.2 TIME SERIES AUGMENTATION METHODS

Various methods for time series augmentation (Demirel & Holz, 2024; Schneider et al., 2024) have been proposed, particularly inspired by image augmentation (Shorten & Khoshgoftaar, 2019), demonstrating excellent performance in time series forecasting tasks. Specifically, Iglesias et al. (2023) proposed a taxonomy of augmentation techniques spanning from simple to advanced approaches. Simple transformation methods involve time, frequency, and magnitude domain augmentation techniques such as slicing (Cao et al., 2020), frequency warping (Cui et al., 2015), and jittering (Flores et al., 2021). Additionally, advanced generative models have been specifically designed to generate realistic time series data, including the GAN-based approaches (such as TimeGAN (Yoon et al., 2019) and Conditional Sig-Wasserstein GAN (Liao et al., 2020)), the VAE-based approaches (such as Conditional VAE (Sohn et al., 2015) and Smoothness-Including Sequential VAE (Li et al., 2020)), and Diffusion-based approaches (Huang et al., 2023). The generated samples can be used for further training of the forecasting models. The third category of data augmentation techniques for time series is the reinforcement learning-based methods (Ko & Ok, 2022; Fu et al., 2022), dynamically scheduling data augmentation operations during reinforcement learning. It is important to note that the first category of methods employs fixed data transformation functions, which may introduce significant noise. While the second and third categories are learning-based, they fail to analyze the characteristics that augmented data should possess or to identify the most effective parts of the data to serve as anchor points for augmentation.

## 3 PRELIMINARY FINDINGS

Following previous work (Wu et al., 2021; Nie et al., 2023a; Liu et al., 2024), we initially conducted explorations on several classical real-world multivariate time series datasets, including ETT (4 subsets), Weather and Electricity. A more detailed description of the datasets can be found in the supplementary materials. To evaluate the impact of varying training data on model performance, we conduct the following experiments. We hypothesize that the model zoo can thoroughly analyze the patterns within each sample. To illustrate the intuition behind our proposed method, we pretrain a model zoo that includes iTransformer, Transformer, PatchTST, and LTSF-Linear, and then backtest them on the same training dataset. By calculating the mean squared error (MSE) for each model, we

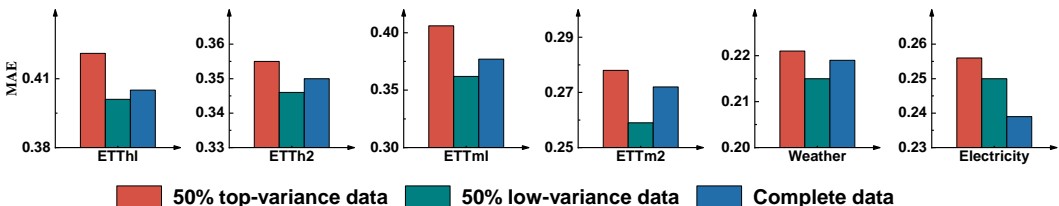

Figure 2: The comparison of the performance of the state-of-the-art forecasting model, iTransformer, on the same test set after training with different datasets. We divide the original training set based on the variance of the prediction error across the pretrained model zoo.

sort the variance of MSE across different models. We then divide the training data into two subsets based on variance: (A) the top $50\%$ of high-variance data and (B) the top $50\%$ of low-variance data, and compare the test results. We train the forecasting model separately using data from groups A and B. Given its superior average performance across the model zoo, we choose iTransformer as the preferred model for these experiments.

The forecasting results are presented in Figure 2, where a lower MAE indicates more accurate predictions. Our experiments reveal that the choice of sub-training sets significantly impacts the training outcomes. Predictions made using group B as training data consistently outperform those using group A across all datasets, and in some instances, even surpass the performance achieved with the full training set. This novel finding suggests that marginal samples, which typically induce high variance in model zoo evaluations, are critical factors that can negatively affect model performance. As a result, we hypothesize that augmenting these specific samples is a more effective strategy.

Therefore, our data augmentation methods are exclusively focused on these marginal samples. By applying learnable data augmentation to enhance these samples, we aim to help the model better capture their patterns, ultimately leading to improved performance.

## 4 METHOD

In this section, we first present the overall training pipeline of our data augmentation method (Section 4.1). We then introduce the neural network architecture as the data generator (Section 4.2) and the reinforcement learning algorithm that optimizes this model (Section 4.3).

### 4.1 OVERALL TRAINING PIPELINE

Inspired by the preliminary results, we can broadly categorize the training data into three groups: easy, hard, and marginal samples. We can partially identify these groups by analyzing the variance in prediction errors across a batch of pretrained forecasting models. The idea is to handle each group individually during the augmentation process. For easy samples, we do not perform augmentation, because repeated playback of data with high similarity and strong regularity does not promote the model to fit more complex samples. For marginal samples, due to the varying ability of different models to learn specific data patterns, only a subset of models can effectively capture these patterns and produce accurate results. In this case, predictions from the model zoo tend to show higher variance. For hard samples, most models consistently make biased predictions with large mean errors but relatively low variance. This can be caused by strong noises or outlier distributions within the dataset, making these samples less suitable for data augmentation.

By leveraging the model zoo as a criterion to identify marginal samples——where model performances diverge——we select these marginal samples as anchor points for data augmentation. As a result, our augmentation efforts will focus primarily on the samples with higher variance in the model zoo. Subsequently, the entire training pipeline consists of three stages:

- Stage A: Train a probabilistic generative model to initialize the neural augmentor, using marginal data and minimizing the reconstruction loss. This model is implemented as a *variational masked autoencoder*, which takes partially masked time series data as input and generates complete data.
- Stage B: Perform a REINFORCE algorithm (Williams, 1992) to enable the neural augmentor to generate data beyond merely replicating the original data distribution.

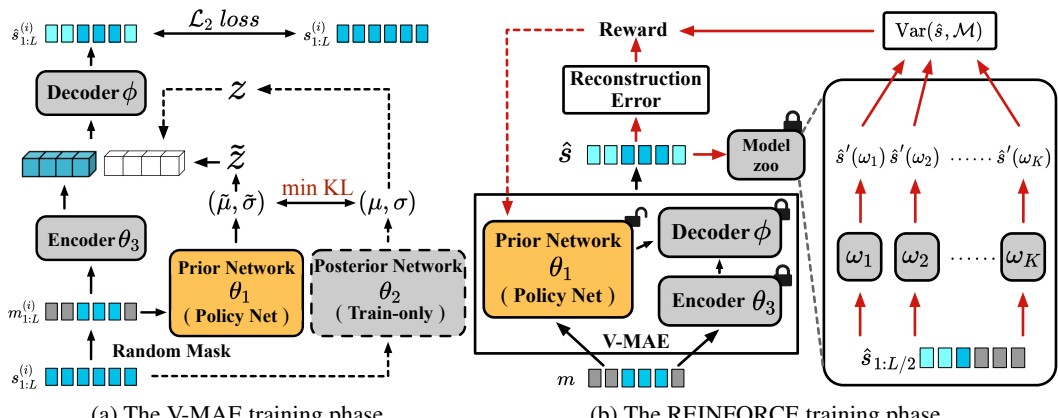

Figure 3: An illustration of the training phases of the neural augmentor. During the V-MAE training phase (a), we optimize the entire probabilistic generative model to approximate the original distribution of the marginal data. In REINFORCE (b), we finetune the prior network in V-MAE, treating it as the policy network, while keeping the parameters of the other V-MAE modules fixed.

- Stage C: Train the forecasting model using both the original data and the augmented data. Specifically, we augment the top $50\%$ samples with the highest variance over the model zoo, tripling the size of the original training set.

## 4.2 Variational Masked Autoencoder

By using a probabilistic generative model as the neural augmentor, denoted as $\hat{s}_{1:L}^{(i)} \sim G(s_{1:L}^{(i)}, z^{(i)})$, we can generate an infinite amount of data by learning a transformation function based on the marginal data $s_{1:L}^{(i)}$, where $i$ is the data index and $L$ is the length of the data sequence. The key is to learn an appropriate distribution of the latent variable $z$, balancing the diversity of the augmented data with its similarity to the original data. For simplicity, we omit the data index in the following notations. The initial learning step involves optimizing $G$ with a data reconstruction objective, minimizing the divergence between the masked data and the original data distribution.

We design a Variational Masked Autoencoder (V-MAE), as illustrated in Figure 3a, which takes masked time series data $m_{1:L}$ as input and outputs the complete corresponding data. The entire architecture contains four modules, including (i) the prior network, which learns the prior distribution of $z$ based on the masked data, (ii) the posterior module, which learns the posterior distribution of $z$ based on the original data, (iii) the data encoder, which extracts significant features from $m_{1:L}$, (iv) the decoder, which generates data from the encoding features and the latent variables. These modules are parametrized by $\theta_{1:3}$ and $\phi$, respectively. We have

$$\begin{aligned} \text{Prior:} \quad & \tilde{z} \sim p(m_{1:L}; \theta_1), \\ \text{Posterior:} \quad & z \sim q(s_{1:L}; \theta_2), \\ \text{Encoding \& Decoding:} \quad & \hat{s}_{1:L} = \text{Dec}\left(\text{concat}\left(\text{Enc}(m_{1:L}; \theta_3), z\right); \phi\right). \end{aligned} \tag{1}$$

We draw the latent variables, which control the diversity of the generated data, from parametrized Gaussian distributions. The mean and standard deviation of these distributions are produced by the prior and posterior modules. We minimize the Kullback–Leibler (KL) divergence between the prior and posterior distributions. After this training stage, we replace the posterior $z$ with the prior $\tilde{z}$ as input to the decoder. The overall objective function is

$$\mathcal{L} = \mathbb{E}_{s \sim D_s} \|\hat{s}_{1:L} - s_{1:L}\|_2^2 + \beta\, \mathcal{D}_{KL}\left(q(z \mid s_{1:L}) \,\|\, p(\tilde{z} \mid m_{1:L})\right), \tag{2}$$

where $D_s$ is the set of the marginal data obtained by measuring the prediction diversity over the model zoo. In line with previous work, we use the $\mathcal{L}_2$ loss to measure the reconstruction error. Notably, the posterior module is used exclusively to constrain the prior learner and is not utilized in subsequent training stages.

In general, this V-MAE design can be integrated into any encoder-decoder-based time series forecasting architecture. In this work, we specifically adopt the encoder and decoder (implemented as a linear projector) from iTransformer (Liu et al., 2024) as the feature extraction and decoding modules.

### 4.3 REINFORCE WITH MODEL ZOO

Our fundamental idea is that merely approximating the distribution of the original data is insufficient, as the aforementioned objective function of V-MAE does not provide clear guidance for transforming the marginal data. This approach can be viewed as broadly spreading the data distribution of the marginal samples in the hope that it will fortuitously cover important patterns in the test set, rather than learning a targeted shift for data augmentation.

An interesting finding from the preliminary experiments is that training the forecasting model on samples with higher prediction error variances across the model zoo leads to better performance. The model zoo, denoted by $\mathcal{M}$, consists of $K$ forecasting models with pretrained parameters $\omega_{1:K}$. This suggests a potential strategy for tuning the neural augmentor — *augmenting the dataset with samples that exhibit lower prediction variance across the model zoo*.

However, a practical challenge is that the "model-zoo variance" is not differentiable and thus cannot be directly used to optimize the augmentor through gradient descent. To tackle this problem, we propose to finetune the neural augmentor using a REINFORCE algorithm. In this framework, the latent space generated by the prior module in V-MAE serves as the action space. We incorporate the variance of the prediction error across the model zoo, evaluated on the augmented data $\hat{s}_{1:L}$, as part of the reward function. As illustrated in Figure 3b, we first backtest $\hat{s}_{1:L}$ using each forecasting model within the model zoo, and then calculate the *model-zoo variance* as

$$\mathrm{Var}\left(\hat{s}_{1:L}, \mathcal{M}\right) = \frac{1}{K} \sum_{1}^{K} \left(\hat{s}'_{L/2:L}(\omega_k) - \bar{s}'_{L/2:L}\right)^2,$$

$$\text{s.t.} \quad \bar{s}'_{L/2:L} = \frac{1}{K} \sum_{1}^{K} \|\hat{s}_{L/2:L} - \hat{s}'_{L/2:L}(\omega_k)\|_2^2,$$

(3)

where $\hat{s}'_{L/2:L}(\omega_k)$ represents the prediction results from the $k$-th pretrained forecasting model. Subsequently, we finetune the V-MAE's prior module through REINFORCE, treating the prior learner as the policy network while keeping the other network parameters fixed. The learning objective is to maximize the following reward function:

$$r = \frac{1}{1 + e^{-k \cdot f(\hat{s}_{1:L}, \mathcal{M})}}, \quad \text{s.t.} \quad f(\hat{s}_{1:L}, \mathcal{M}) = \frac{1}{\|\hat{s}_{1:L} - s_{1:L}\|_2^2 \cdot \mathrm{Var}\left(\hat{s}_{1:L}, \mathcal{M}\right)}. \quad (4)$$

Our goal is to generate new samples that not only resemble the original data but also effectively supplement the training set. This approach encourages that *the transformations of the marginal data can also be marginal data*. In the reward function, a scaled-sigmoid function is employed to minimize the likelihood of rewards clustering around $0$ or $1$ controlled by hyperparameter $k$. This approach ensures that, despite potential order-of-magnitude differences in reconstruction error and backtest variance within the model zoo, the reward function can learn effectively from these variations. Based on this reward function, the optimization of the policy network can be formulated as follows via gradient ascent, where $\alpha$ is the learning rate:

$$\theta_1 \leftarrow \theta_1 + \alpha \cdot r \cdot \nabla_{\theta_1} \log p\left(\tilde{z} \mid m_{1:L}; \theta_1\right). \quad (5)$$

To summarize, by using the REINFORCE algorithm to optimize the neural augmentor, we effectively leverage the model zoo's ability to assess data properties, resulting in the generation of samples with patterns that time series forecasting models can more easily capture.

## 5 EXPERIMENTS

### 5.1 EXPERIMENTAL SETUP

**Datasets.** Following previous work in model zoo (Liu et al., 2024; Nie et al., 2023a; Zeng et al., 2023; Vaswani et al., 2017), we thoroughly evaluate the proposed AutoTSAug on four real-world

Table 1: The performance of iTransformer without vs. with data augmentation. We use the same random seed for training in all three scenarios. For AutoTSAug, since it employs a stochastic neural augmentor, we conduct the training process three times, each with a different set of augmentation data. We report the mean and standard deviation of these experiments.

| Dataset | Original | | GaussianAugmentor | | AutoTSAug | |
|---|---|---|---|---|---|---|
| | MAE | MSE | MAE | MSE | MAE | MSE |
| ETTh1 | 0.405 | 0.387 | 0.407±0.02 | 0.392±0.01 | **0.397±0.02** | **0.382±0.01** |
| ETTh2 | 0.350 | 0.301 | 0.352±0.01 | 0.307±0.01 | **0.347±0.01** | **0.294±0.01** |
| ETTm1 | 0.377 | 0.341 | 0.374±0.02 | 0.340±0.02 | **0.363±0.01** | **0.327±0.01** |
| ETTm2 | 0.272 | 0.186 | 0.272±0.01 | 0.187±0.00 | **0.262±0.01** | **0.179±0.01** |
| Weather | 0.219 | 0.178 | 0.227±0.01 | 0.187±0.01 | **0.207±0.00** | **0.170±0.00** |
| Electricity | 0.239 | 0.148 | 0.243±0.01 | 0.150±0.00 | **0.237±0.01** | **0.147±0.01** |
| Traffic | 0.269 | 0.392 | 0.269±0.01 | 0.394±0.02 | **0.268±0.01** | **0.391±0.01** |

Table 2: The impact of AutoTSAug applied to each model in the model zoo. Specifically, we report the average performance of the models trained with three different augmented datasets.

| Train Data | iTransformer | | Transformer | | PatchTST | | LTSF-Linear | | AVG | |
|---|---|---|---|---|---|---|---|---|---|---|
| | MAE | MSE | MAE | MSE | MAE | MSE | MAE | MSE | MAE | MSE |
| ETTh1 (Orig.) | 0.405 | 0.387 | 0.750 | 0.884 | 0.408 | 0.394 | 0.396 | 0.383 | 0.490 | 0.512 |
| +AutoTSAug | 0.397 | 0.382 | 0.684 | 0.758 | 0.396 | 0.379 | 0.395 | 0.383 | 0.468 | 0.476 |
| Promotion | 1.98% | 1.30% | 8.90% | 14.3% | 2.94% | 3.81% | 0.25% | 0% | **4.49%** | **7.03%** |
| ETTh2 (Orig.) | 0.350 | 0.301 | 0.974 | 1.362 | 0.343 | 0.294 | 0.380 | 0.329 | 0.512 | 0.572 |
| +AutoTSAug | 0.347 | 0.294 | 0.849 | 1.134 | 0.343 | 0.293 | 0.378 | 0.326 | 0.479 | 0.512 |
| Promotion | 0.86% | 2.33% | 12.8% | 16.7% | 0% | 0.34% | 0.53% | 0.91% | **6.45%** | **10.5%** |
| ETTm1 (Orig.) | 0.377 | 0.341 | 0.477 | 0.461 | 0.360 | 0.321 | 0.374 | 0.346 | 0.397 | 0.367 |
| +AutoTSAug | 0.363 | 0.327 | 0.460 | 0.452 | 0.361 | 0.322 | 0.362 | 0.329 | 0.387 | 0.358 |
| Promotion | 3.71% | 4.11% | 3.56% | 1.95% | -0.28% | -0.31% | 3.21% | 4.91% | **2.52%** | **2.45%** |
| ETTm2 (Orig.) | 0.272 | 0.186 | 0.521 | 0.434 | 0.260 | 0.178 | 0.271 | 0.182 | 0.331 | 0.245 |
| +AutoTSAug | 0.262 | 0.179 | 0.501 | 0.411 | 0.259 | 0.177 | 0.264 | 0.180 | 0.322 | 0.237 |
| Promotion | 3.68% | 2.76% | 3.84% | 5.30% | 0.38% | 0.56% | 2.58% | 1.10% | **2.72%** | **3.27%** |
| Weather (Orig.) | 0.219 | 0.178 | 0.464 | 0.409 | 0.217 | 0.175 | 0.255 | 0.197 | 0.289 | 0.240 |
| +AutoTSAug | 0.207 | 0.170 | 0.368 | 0.310 | 0.216 | 0.175 | 0.235 | 0.194 | 0.257 | 0.212 |
| Promotion | 5.48% | 4.49% | 20.69% | 24.21% | 0.46% | 0% | 7.84% | 1.52% | **11.07%** | **11.67%** |
| Elec. (Orig.) | 0.239 | 0.148 | 0.355 | 0.261 | 0.249 | 0.163 | 0.278 | 0.195 | 0.280 | 0.192 |
| +AutoTSAug | 0.237 | 0.147 | 0.350 | 0.257 | 0.251 | 0.165 | 0.274 | 0.193 | 0.277 | 0.191 |
| Promotion | 0.84% | 0.68% | 1.41% | 1.53% | -0.80% | -1.23% | 1.44% | 1.03% | **1.07%** | **0.52%** |
| Traffic (Orig.) | 0.269 | 0.392 | 0.484 | 0.866 | 0.276 | 0.434 | 0.397 | 0.650 | 0.357 | 0.585 |
| +AutoTSAug | 0.268 | 0.391 | 0.466 | 0.837 | 0.276 | 0.433 | 0.387 | 0.647 | 0.349 | 0.577 |
| Promotion | 0.37% | 0.26% | 3.72% | 3.35% | 0% | 0.23% | 2.52% | 0.46% | **2.24%** | **1.37%** |

datasets that are publicly available, including ETT (4 subsets), Traffic, Electricity, Weather. Please refer to the appendix for more details on these datasets.

**Model zoo.** We use a set of pretrained forecasting models, referred to as the model zoo, to establish the diversity criterion for identifying marginal samples and to construct the reward function in the REINFORCE algorithm. The model zoo consists of four models: Transformer (Vaswani et al., 2017), PatchTST (Nie et al., 2023a), LTSF-Linear (Zeng et al., 2023), and iTransformer (Liu et al., 2024). Our experiments demonstrate that AutoTSAug is consistently effective across the model zoo, as it improves the performance of each model when it is used as the final forecasting model.

**Augmentation baseline with Gaussian noises.** Inspired by prior literature (Iglesias et al., 2023), we employ a traditional data augmentor based on simple data transformations. It enhances the diversity of the training data by adjusting the controllable variances and means of the added Gaussian noise. As in previous work, this method applies consistent augmentation across all training data.

Table 3: The correspondence of the final results with how many marginal samples (with high prediction variance across the model zoo) are used as the augmentation anchor points in AutoTSAug.

| Augmented Data | ETTh1 | | ETTh2 | | ETTm1 | | ETTm2 | | Weather | |
|---|---|---|---|---|---|---|---|---|---|---|
| | MAE | MSE | MAE | MSE | MAE | MSE | MAE | MSE | MAE | MSE |
| 30% Top-Variance Data | 0.400 | 0.383 | **0.346** | 0.295 | 0.369 | 0.343 | 0.265 | 0.184 | **0.207** | **0.170** |
| 50% Top-Variance Data | **0.397** | **0.382** | 0.347 | **0.294** | **0.363** | **0.327** | **0.262** | **0.179** | 0.209 | 0.173 |
| All Training Data | 0.408 | 0.390 | 0.351 | 0.304 | 0.372 | 0.337 | 0.271 | 0.186 | 0.224 | 0.185 |

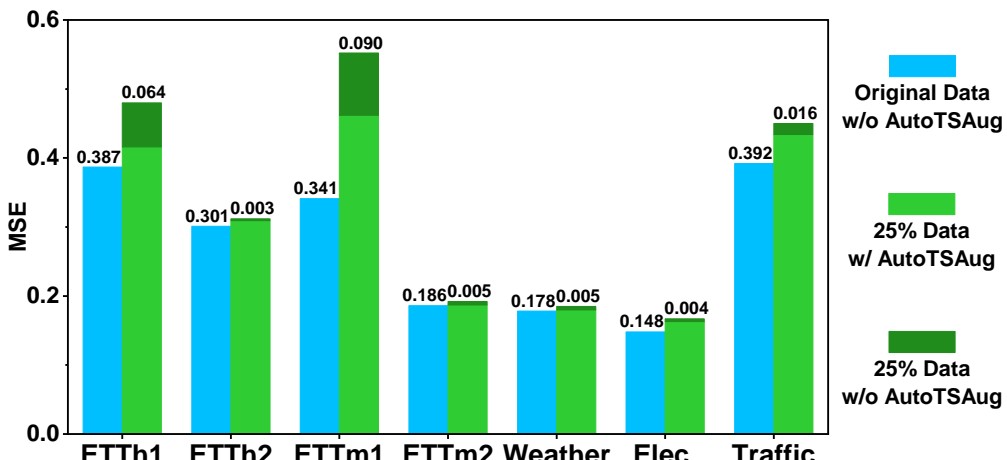

Figure 4: Few-shot experiments: We compare the results on a partial training set (*green*) and the full training set (*blue*). After applying AutoTSAug, the model's performance on the 25% subset is significantly close to the performance on the full dataset.

## 5.2 Forecasting Results

**Main results.** We employ a fixed lookback length of 96 timestamps across all datasets and report the multivariate sequence prediction results with prediction lengths of 96 timestamps. Under this setting, we find that iTransformer outperforms other models on average. Consequently, we use iTransformer as the primary forecasting model and compare the results of training iTransformer without and with different augmentation methods. It is noteworthy that both the traditional Gaussian augmentor and AutoTSAug augment the original training set threefold. In most cases in Table 1, the inclusion of random noise data negatively impacts training. In contrast, data augmented by AutoTSAug consistently results in better testing performance across all datasets.

**Consistent improvements upon other forecasting models.** In addition to using iTransformer as the default forecasting model in Stage C, we further investigate the potential benefits of AutoTSAug for improving other models from the model zoo. As illustrated in Table 2, AutoTSAug consistently improves the prediction accuracy across various forecasting models.

**Few-shot experiments.** Effective data augmentation methods should be able to deal with the training difficulty due to insufficient training data. To explore this, we conduct few-shot experiments using 25% of the original training data. As shown in Figure 4, limited training data significantly reduces the prediction accuracy of the model. However, after applying AutoTSAug, the performance of the model is significantly closer to the performance when we use the full dataset.

## 5.3 Model Analyses

**Shall we augment all training samples?** Inspired by preliminary findings, we only augment the training data with higher variance in the model zoo to enhance the model's learning from marginal samples. Specifically, we augment the top 50% periods with the highest variance. What would be the impact of augmenting more or fewer samples? To investigate this, we conduct the following experiments: augmenting the top 30% high-variance samples, the top 50% high-variance samples,

Table 4: Comparison of the impact of the REINFORCE algorithm on data augmentation. We similarly use augmented data which is three times the amount of the original data to train iTransformer. By comparing the test results, we assess the quality of the augmented data.

| REINF. | ETTH1 MAE | ETTH1 MSE | ETTH2 MAE | ETTH2 MSE | ETTM1 MAE | ETTM1 MSE | ETTM2 MAE | ETTM2 MSE | WEATHER MAE | WEATHER MSE | ELEC. MAE | ELEC. MSE | TRAFFIC MAE | TRAFFIC MSE |
|---|---|---|---|---|---|---|---|---|---|---|---|---|---|---|
| × | 0.405 | 0.386 | 0.349 | 0.300 | 0.377 | 0.341 | 0.272 | 0.186 | 0.221 | 0.182 | 0.240 | 0.149 | 0.269 | 0.393 |
| ✓ | **0.397** | **0.382** | **0.347** | **0.294** | **0.363** | **0.327** | **0.262** | **0.179** | **0.207** | **0.170** | **0.237** | **0.147** | **0.268** | **0.391** |

Table 5: Computational cost for each training stage. The total training time required for dataset augmentation, including Stage A and Stage B, is notably shorter than the time needed for training the forecasting models, indicating our method's reasonable efficiency.

| DATASET | STAGE A: V-MAE | STAGE B: REINFORCE | STAGE C: FORECASTING iTRANSFORMER | STAGE C: FORECASTING PATCHTST |
|---|---|---|---|---|
| ETTH1 | 2MIN | 4MIN | 9MIN | 14MIN |
| ELEC. | 1H 41MIN | 2H 48MIN | 6H 37MIN | 10H 04MIN |
| TRAFFIC | 4H 42MIN | 7H 42MIN | 17H 43MIN | 27H 13MIN |

and the entire training set. For consistency, the total amount of augmented data was kept constant at three times the original training set and applied to iTransformer. As shown in Table 3, augmenting the entire dataset directly was less effective than data-dependent augmentation, which is in line with our preliminary findings. Furthermore, augmenting different percentages of high-variance samples yields varying degrees of improvement, demonstrating that our data augmentation model can adaptively enhance marginal samples, thereby improving the performance of the forecasting model.

**Ablation study on REINFORCE.** The design of REINFORCE enables our model to optimize the distribution of the latent variable $z$ based on the backtest results across the model zoo, thereby generating augmented data that balances data diversity and similarity to the original data. Intuitively, this design is inspired by preliminary findings. However, V-MAE can also serve as a complete model before undergoing REINFORCE training. To investigate the significance of the REINFORCE method, we train the iTransformer model using augmented data generated by V-MAE without REINFORCE optimization. As shown in Table 4, REINFORCE improves the prediction accuracy on multiple datasets, demonstrating the significance of the module to the quality of the augmented data.

**Computational cost.** The proposed data augmentation method involves training V-MAE and employing the REINFORCE algorithm, and it requires backtesting across multiple models within the model zoo. These processes introduce additional computational overhead compared to basic prediction models. As shown in Table 5, we report the training time for V-MAE (Stage A), REINFORCE (Stage B), and the forecasting model (Stage C). The increased training time (including Stage A and Stage B) is considered acceptable due to the significant performance gains. Besides, it is notably shorter than the time needed for training the forecasting models. We evaluate the computational cost using a single NVIDIA 3090 GPU.

# 6 CONCLUSIONS AND LIMITATIONS

In this paper, we proposed AutoTSAug, a novel data augmentation method driven by reinforcement learning. There are two contributions in methodology: First, our method automatically identifies the critical training data for augmentation, termed the marginal samples, based on prediction diversity from a set of pretrained forecasting models. Second, AutoTSAug exploits a variational masked autoencoder in conjunction with the REINFORCE algorithm to generate new data from these marginal samples. AutoTSAug significantly boosts forecasting performance while maintaining minimal computational overhead by leveraging a learnable policy to transform the marginal samples.

One unresolved issue in this study is the need for multiple pretrained models. The presence of models with poor performance could potentially impact the quality of the generated data. To address this, we incorporate recent competitive models in the model zoo. Future work will focus on enhancing the robustness of our approach to effectively adapt to an imperfect model zoo.

ETHICS STATEMENT

In this work, we adhere to the highest ethical standards across all stages of research. No human subjects were involved, and no personal data was used, ensuring compliance with privacy and security protocols. All datasets utilized are publicly available, mitigating concerns related to sensitive information exposure. We acknowledge the potential for forecasting models to generate harmful insights if misapplied; therefore, we encourage careful consideration of the context and application domain when deploying these models.

REPRODUCIBILITY STATEMENT

We prioritize the reproducibility of our work. All results can be reproduced by following the experimental details presented in Section 5.1 and Appendix A.2. Additionally, we provide the source code in the supplementary materials.

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

# A APPENDIX

## A.1 OVERALL TRAINING PIPELINE

We present the pseudocode of the overall training pipeline in Algorithm 1.

---

**Algorithm 1** Overall training pipeline

---

1: **Given:** Time series samples from training set $s_{1:L}^{(i)}$
2: **Key problem:** Which samples should be augmented and how to augment them?
3: // Stage A: Train the V-MAE supervised by original data
4: V-MAE can be parameterized as $\theta_1, \theta_2, \theta_3, \phi$, all parameters are optimized during the training phase.
5: // Stage B: Data filtering by model zoo variance
6: Pretrain a model zoo and assess on the training set
7: The top 50% samples with large variance on model zoo are found from the training set and formulated as $s_{1:L}$
8: // REINFORCE with model zoo
9: Fixed parameters $\theta_2, \theta_3, \phi$
10: **while** not converged **do**
11:     Sample batch of time series samples and mask randomly, formulated as $m_{1:L}$
12:     Input masked data $m_{1:L}$ to pretrained V-MAE, calculate the reward $r$ by the output of V-MAE and pretrained model zoo.
13:     Update the policy net parameters: $\theta_1 \leftarrow \theta_1 + \alpha \cdot r \cdot \nabla_{\theta_1} \log E_{\theta_1}(\tilde{\mathbf{z}} \mid m_{1:L})$
14: **end while**
15: Stage C: Train the forecasting model
16: Generate augmented data by AutoTSAug
17: Further training the forecasting model (e.g., iTransformer) by augmented data

---

## A.2 IMPLEMENTATION DETAILS

### A.2.1 DETAILS OF THE DATASETS

Here is a detailed description of the four experiment datasets: (1) ETT (Zhou et al., 2021) consists of two hourly-level datasets (ETTh) and two 15-minute-level datasets (ETTm). Each of them contains 7 factors of electricity transformers including load and oil temperature from July 2016 to July 2018. (2) Traffic (Wu et al., 2021) is a collection of road occupancy rates measured by 862 sensors on San Francisco Bay area freeways from January 2015 to December 2016. (3) ECL (Wu et al., 2021) collects hourly electricity consumption of 321 clients from 2012 to 2014. (4) Weather (Wu et al., 2021) includes 21 meteorological indicators, such as air temperature and humidity, recorded 10 minutes from the weather station of the Max Planck Biogeochemistry Institute in 2020.

We follow the data processing method of iTransformer (Liu et al., 2024), dividing the dataset into training, validation, and test sets, with this partitioning strictly aligned in chronological order. In Table 6, we provide the number of variables (*i.e.*, the feature dimension at a single time point) in each dataset, the total number of time points, and the number of time points within each set of the train-validation-test partitions.

Table 6: Details of the datasets. *Features* denotes the number of data variables in each dataset. *Time points* refers to the total number of time points in the dataset. *Partition* indicates the number of time points allocated to each subset in the (train, validation, test) splits.

| | ETTh1/ETTh2 | ETTm1/ETTm2 | Traffic | Electricity | Weather |
|---|---|---|---|---|---|
| Features | 7 | 7 | 862 | 321 | 21 |
| Time points | 14307 | 57507 | 17451 | 26211 | 52603 |
| Partition | (8545, 2881, 2881) | (34465, 11521, 11521) | (12185, 1757, 3509) | (18317, 2633, 5261) | (36792, 5271, 10540) |

### A.2.2   HYPERPARAMETERS AND COMPUTING RESOURCES

In Table 7, we provide the hyperparameter details of V-MAE and REINFORCE. For the encoder and decoder, we adopt the identical hyperparameters as those employed in iTransformer. We perform all experiments on an NVIDIA RTX 3090 GPU.

Table 7: Hyperparameters of AutoTSAug.

| Notation | Hyperparameter | Description |
|----------|----------------|-------------|
| $\alpha$ | 0.001 | Learning rate of REINFORCE |
| $\beta$ | 0.1 | Weight of KL-divergence in the V-MAE loss function |
| $L$ | 96 | Time series periods length |
| $k$ | 100 | Parameters of scaled sigmoid |
| $N$ | 32 | Batch size for V-MAE training |

