# OpenReview forum: "Learning Augmentation Policies from A Model Zoo for Time Series Forecasting"
_ICLR.cc/2025/Conference — ICLR 2025 Conference Withdrawn Submission_

### Official Review · Reviewer_Ec4E · 2024-11-03

**Soundness:** 4
**Presentation:** 4
**Contribution:** 2
**Rating:** 5
**Confidence:** 4

**Summary:**

The paper proposes a data augmentation method for time series prediction. The paper focuses on augmenting "marginal samples" which are samples that has high prediction variance across a variety of different prediction models. The paper then propose a generative approach using V-MAEs to augment the marginal samples and to train the generative model via a reinforcement learning approach.

**Strengths:**

- The paper proposes a interesting perspective on data augmentation by focusing entirely on high prediction variance samples.
- The paper is generally well written and the methods and results are presented well.

**Weaknesses:**

- In the Related Works section, the paper presents other generative based and RL-based methods for time-series data augementation yet only compares their proposed method with the Gaussian noise augmentor. It would be nice to see more baseline comparisons (at least one  each from generative based and RL-based methods).
- It is not super convincing that augmenting only marginal samples results in consistantly significant improvements for the prediction model, as most of the results presented in Table 2 having  a <5% improvement with the augmented data, with the biggest improvement coming from the basic Transformer model.
- Moreover, it is not entirely clear that AutoTSAug is able to consistantly morph the marginal samples into samples that exhibit lower prediction variance in the model zoo.

**Questions:**

- Are the marginal samples consistant across different training instances of the same model? E.g. if the model zoo is initialized with different parameters or trained with different hyperparameters does it affect which samples are considered marginal?
- Since the paper proposes an RL based training approach, does the proposed method use a multi-step training approach where the initial recontruction is fed into the encoder as the new "state" and policy model would then further augment the sample. Or do the proposed method use a single step approach and if so is that enough to significantly modify the samples towards the reward function?

---

### Official Review · Reviewer_CgNu · 2024-11-04

**Soundness:** 2
**Presentation:** 3
**Contribution:** 3
**Rating:** 5
**Confidence:** 4

**Summary:**

The paper presents AutoTSAug, a novel data augmentation method for time series forecasting that uses reinforcement learning to learn optimal augmentation policies. The key innovations are: (1) using a ‘‘model zoo’’ of pretrained forecasting models to identify ‘‘marginal samples’’ that would benefit most from augmentation, and (2) employing a variational masked autoencoder trained with REINFORCE to generate augmented data that reduces prediction variance across the model zoo. The method shows consistent improvements over baselines across multiple datasets.

**Strengths:**

- The use of model zoo diversity to identify samples for augmentation.
- Interesting combination of VAE and RL.
- Comprehensive empirical validation across multiple datasets.
- Thorough ablation studies supporting design choices.
- Practically useful with reasonable computational overhead.

**Weaknesses:**

- Limited theoretical justification for focusing exclusively on high-variance samples. Could you provide a more formal theoretical justification of this claim? Or empirically prove that augmenting low variance samples (hard ones) is not beneficial?
- Potential oversight of valuable transformations for low-variance samples. Could you apply your augmentation framework to augment hard samples, with a different reward function that would improve the forecasting results on these samples?
- Over-reliance on model zoo’s diversity criterion without stability analysis.
- Risk of generating uniformly poor samples due to variance-based reward. Are there any safeguards to prevent generating uniformly poor samples (e.g. variance >> generation error)?
- Limited comparison with state-of-the-art augmentation methods.
- Missing analysis of hard samples that consistently perform poorly.
- Insufficient justification for choosing REINFORCE over other RL algorithms.

**Questions:**

1) How sensitive is the method to the choice of models in the zoo? What criteria should be used for model selection?
2) Why not consider augmenting ‘‘hard samples’’ that consistently perform poorly across the model zoo? Perhaps, augmenting these hard samples may help the models uncover their underlying patterns, and improve their performance?
3) How does the approach ensure that minimizing variance doesn’t lead to uniformly poor samples? For instance, if the variance is very high, and the agent gives more importance to the variance criterion.
4) What advantages does REINFORCE offer over alternative policy optimization methods like PPO or TRPO?
5) How does the method compare to approaches with theoretical guarantees like Recursive Time Series Data Augmentation (RIM)?

---

### Official Review · Reviewer_kFJn · 2024-11-04

**Soundness:** 2
**Presentation:** 2
**Contribution:** 2
**Rating:** 5
**Confidence:** 4

**Summary:**

This work introduces a novel data augmentation method, AutoTSAug, that transforms high-diversity or marginal samples in time series data to better align with standard patterns, using reinforcement learning to guide the transformations. By leveraging a model zoo, the method identifies challenging samples with high prediction variance and applies a variational masked autoencoder to generate augmented, normalized versions of these samples. This approach aims to reduce prediction error variance and improve model stability by effectively normalizing outliers rather than removing them.

**Strengths:**

Normalizing challenging, high-diversity samples rather than removing them is a notable departure from traditional outlier detection and data cleaning practices. The use of reinforcement learning to guide this normalization process adds further novelty, as it enables the model to learn an optimal augmentation policy that reduces prediction variance across a model zoo.

**Weaknesses:**

A key weakness of the paper lies in its lack of engagement with the outlier detection and data cleaning literature, which limits the reader's ability to understand the contribution in context.

**Questions:**

I am not sure why there is no discussion of outlier and out-of-distribution detection or concepts introduced in the data cleaning literature. I think it is crucial to see the contribution and position of this work in those literatures and provide discussions for the proposed method against the approaches in those literatures.

---

### Official Review · Reviewer_TrPt · 2024-11-04

**Soundness:** 3
**Presentation:** 3
**Contribution:** 2
**Rating:** 3
**Confidence:** 4

**Summary:**

The paper introduces a novel data augmentation method for time series forecasting. More specifically, the proposed method
leverages a model zoo of pretrained forecasting models to identify the so called"marginal samples" - training instances where models show a high prediction diversity. Focusing augmentation on these marginal samples is more effective than uniform augmentation across all data. To learn the augmentation policy the method uses a variational masked autoencoder (V-MAE) as the base augmentation model. They applies REINFORCE algorithm to optimise the augmentation policy using model zoo prediction variance as feedback. The goal is to generate augmented data that reduces prediction variance across models.

**Strengths:**

Presents novel ideas for data augmentation. Rather than treating all data equally for augmentation, introduces the concept of identifying "marginal samples" that would benefit most from augmentation.
A comprehensive empirical validation across multiple datasets and models and a thorough ablation studies examining key components is presented.
Overall well-structured presentation progressing from motivation to implementation.
The design choices are well motivated.

**Weaknesses:**

My main concern with this paper is that modest gains in results don't seem to justify the expensive and complicated method proposed. The requirement of multiple pre trained models itself is quite expensive. The training of the augmentation policy seems quite compute intensive. The performance gains are marginal and are primarily driven by the base transformer. On modern transformer based forecasting methods such as such as patchtst and itransformer the gains are marginal and it even underperforms in some cases. Overall, I think the performance gains don't justify the computation expenses required.

**Questions:**

Please see the weaknesses.

---

### Note · Authors · 2024-11-13

I have read and agree with the venue's withdrawal policy on behalf of myself and my co-authors.